# Using Predictive and Differential Methods with K²-Raster Compact Data Structure for Hyperspectral Image Lossless Compression †

**Kevin Chow *** , **Dion Eustathios Olivier Tzamarias, Ian Blanes and Joan Serra-Sagristà**

Department of Information and Communications Engineering, Universitat Autònoma de Barcelona, 08193 Cerdanyola del Vallès, Barcelona, Spain; dion.tzamarias@uab.cat (D.E.O.T.); ian.blanes@uab.cat (I.B.); joan.serra@uab.cat (J.S.-S.)

* Correspondence: kevin.chow@uab.cat

† This paper is an extended version of our paper publihsed in the 6th ESA/CNES International Workshop on On-Board Payload Data Compression Proceedings.

**Abstract:** This paper proposes a lossless coder for real-time processing and compression of hyperspectral images. After applying either a predictor or a differential encoder to reduce the bit rate of an image by exploiting the close similarity in pixels between neighboring bands, it uses a compact data structure called $k^2$-raster to further reduce the bit rate. The advantage of using such a data structure is its compactness, with a size that is comparable to that produced by some classical compression algorithms and yet still providing direct access to its content for query without any need for full decompression. Experiments show that using $k^2$-raster alone already achieves much lower rates (up to 55% reduction), and with preprocessing, the rates are further reduced up to 64%. Finally, we provide experimental results that show that the predictor is able to produce higher rates reduction than differential encoding.

**Keywords:** compact data structure; quadtree; $k^2$-tree; $k^2$-raster; DACs; 3D-CALIC; M-CALIC; hyperspectral images

## 1. Introduction

Compact data structures [1] are examined in this paper as they can provide real-time processing and compression of remote sensing images. These structures are stored in reduced space in a compact form. Functions can be used to access and query each datum or groups of data directly in an efficient manner without an initial full decompression. This compact data should also have a size which is close to the information-theoretic minimum. The idea was explored and examined by Guy Jacobson in his doctoral thesis in 1988 [2] and in a paper published by him a year later [3]. Prior to this, works had been done to express similar ideas. However, Jacobson's paper is often considered the starting point of this topic. Since then it has gained more attention and a number of research papers have been published. Research on algorithms such as FM-index [4,5] and Burrows-Wheeler transform [6] were proposed and applications were released, notable examples of which include bzip2 (https://linux.die.net/man/1/bzip2), Bowtie [7] and SOAP2 [8]. One of the advantages of using compact data structures is that the compressed data form can be loaded into main memory and accessed directly. The smaller compressed size also helps data move through communication channels faster. The other advantage is that there is no need to compress and decompress the data as is the case with data compressed by a classical compression algorithm such as gzip or bzip2, or by a specialized algorithm such as CCSDS 123.0-B-1 [9] or KLT+JPEG 2000 [10,11]. The resulting image will have the same quality as the original.

Hyperspectral images are image data that contain a multiple number of bands from across the electromagnetic spectrum. They are usually taken by hyperspectral satellite and airborne sensors. Data are extracted from certain bands in the spectrum to help us find the objects that we are specifically looking for, such as oil fields and minerals. However, due to their large sizes and the huge amount of data that have been collected, hyperspectral images are normally compressed by lossy and lossless algorithms to save space. In the past several decades, a lot of research studies have gone into keeping the storage sizes to a minimum. However, to retrieve the data, it is still necessary to decompress all the data. With our approach using compact data structures, we can query the data without fully decompressing them in the first place, and this is the main motivation for this work.

Prediction is one of the schemes used in lossless compression. CALIC (Context Adaptive Lossless Image Compression) [12,13] and 3D-CALIC [14] belong to this class of scheme. In 1994, Wu et al. introduced CALIC, which uses both context and prediction of the pixel values. In 2000, the same authors proposed a related scheme called 3D-CALIC in which the predictor was extended to the pixels between bands. Later in 2004, Magli et al. [15] proposed M-CALIC whose algorithm is related to 3D-CALIC. All these methods take advantage of the fact that in a hyperspectral image, neighboring pixels in the same band (spatial correlation) are usually close to each other and even more so for neighboring pixels of two neighboring bands (spectral correlation).

Differential encoding is another way of encoding an image by taking the difference between neighboring pixels and in this work, it is a special case of the predictive method. It only takes advantage of the spectral correlation. However, this correlation between the pixels in the bands will become smaller as the distance between the bands are further apart and therefore, its effectiveness is expected to decrease when the bands are far from each other.

The latest studies on hyperspectral image compression, both lossy and lossless, are focused on CCSDS 123.0, vector quantization, Principal Component Analysis (PCA), JPEG2000, and Lossy Compression Algorithm for Hyperspectral Image Systems (HyperLCA), among many others. Some of these research works are listed in [16–19]. In this work, however, we investigate lossless compression of hyperspectral images through the proposed $k^2$-raster for 3D images, which is a compact data structure that can provide bit-rate reduction as well as direct access to the data without full decompression. We also explore the use of a predictor and a differential encoder as preprocessing on the compact data structure to see if it can provide us with further bit-rate reduction. The predictive method and the differential method are also compared. The flow chart shown in Figure 1 depicts how the encoding/decoding of this proposal works.

This paper is organized as follows: In Section 2, we present the $k^2$-raster and discuss it in detail, beginning with quadtree, followed by $k^2$-tree and $k^2$-raster. Later in the same section, details of the predictive method and the differential method are discussed. Section 3 shows the experimental results on how the two methods fare using $k^2$-raster on hyperspectral images, and more results on how some other factors such as using different $k$-values can affect the bit rates. Finally, we present our conclusions in Section 4.

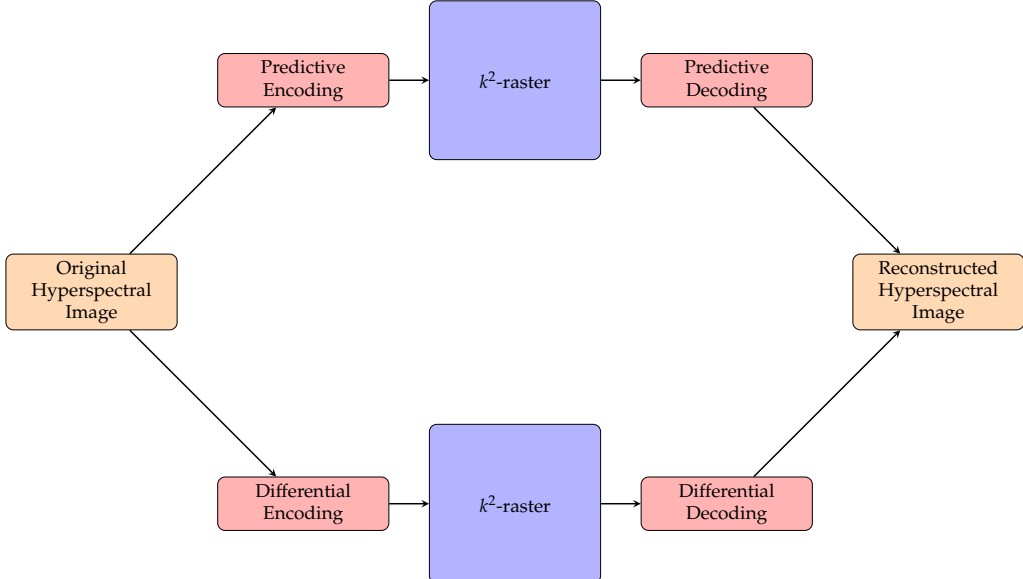

**Figure 1.** A flow chart showing the encoding and decoding of this coder.

## 2. Materials and Methods

One way to build a structure that is small and compact is to make use of a tree structure and do it without using pointers. Pointers usually take up a large amount of space, with each one having a size in the order of 32 or 64 bits for most modern-day machines or programs. A tree structure with n pointers will have a storage complexity of $\mathcal{O}(n \log n)$ whereas a pointer-less tree only occupies $\mathcal{O}(n)$. For pointer-less trees, to get at the elements of the structure, rank and select functions [3] are used, and that only requires simple arithmetic to find the parent's and child's positions. This is the premise that compact data structures are based on. In this work, we will use $k^2$-raster from Ladra et al. [20], a concept which was developed from $k^2$-tree, also a type of compact data structure, as well as the idea of using recursive decomposition of quadtrees. The results of $k^2$-raster were quite favorable for the data sets that were used. Therefore, we are extending their approach for hyperspectral images and investigate whether it would be possible to use that structure for 3D hyperspectral images. The Results section will show us that the results are quite competitive compared to other commonly-used classical compression techniques. There is a bit-rate reduction of up to 55% for the testing images. Upon more experimentation with predictive and differential preprocessing, a further bit-rate reduction of up to 64% can be attained. For that reason, we are proposing in this paper our encoder using the predictor or differential method on $k^2$-raster for hyperspectral images.

### 2.1. Quadtrees

Quadtree structures [21], which have been used in many kinds of data representations such as image processing and computer graphics, are based on the principle of recursive decomposition. As there are many variants of quadtree, we will describe the one that is pertinent to our discussion: region quadtree. Basically, a quadtree is a tree structure where each internal node has 4 children. Given a 2D square matrix, it is partitioned recursively into four equal subquadrants. If a tree is built to represent this, it will have a root node at level 0 with 4 children nodes at level 1, each child representing a node and a subquadrant. Next, if the subquadrant has a size larger than $2^2$, then each of these subquadrants will be partitioned to give 4 more children and a new level 2 is added to the tree. Note that the tree nodes are traversed in a left to right order.

Considering a matrix of size $n \times n$ where $n$ is a power of 2, it is recursively divided until each subquadrant has a size of $2^2$. For example, if the size of the matrix is 8 × 8, after the recursive division of matrix, $(8^2)/(2^2) = 16$ subquadrants are obtained. It should be noted that the value of $n$ in the image matrix needs to be a power of 2. Otherwise, the matrix has to be enlarged widthwise and heightwise to

a value which is the next power of 2, and these additional pixels will be padded with zeros. As $k^2$-trees are based on quadtrees, the division and the resulting tree of a quadtree are very similar to those of a $k^2$-tree. Figure 2 illustrates how a quadtree's recursive partitioning works.

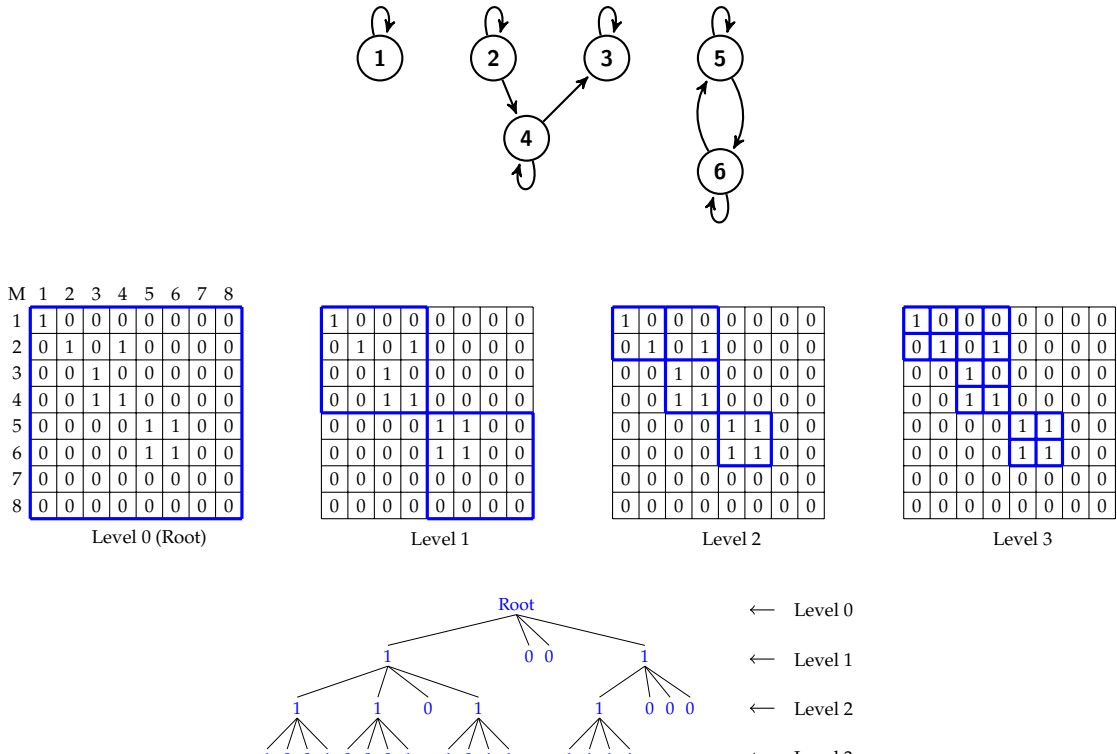

**Figure 2.** A graph of 6 nodes (top) with its $8 \times 8$ binary adjacency matrix at various stages of recursive partitioning. At the bottom, a $k^2$-trees ($k$=2) is constructed from the matrix.

## 2.2. LOUDS

$k^2$-tree is based on unary encoding and LOUDS, which is a compact data structure introduced by Guy Jacobson in his paper and thesis [2,3]. A bit string is formed by a breadth-first traversal (going from left to right) of an ordinal (rooted, ordered) tree structure. Each parent node is encoded with a string of '1' bits whose length indicates the number of children it has and each string ends with a '0' bit. If the parent node has no children, only a single '0' bit suffices.

The parent and child relationship can be computed by two cornerstone functions for compact data structures: rank and select. These functions give us information about the node's first-child, next-sibling(s), and parent, without the need of using pointers. They are described below:

| | |
|---|---|
| $\text{rank}_b(m)$ | returns the number of bits which are set to $b$, left of position $m$ (inclusive) in the bitmap where $b$ is 0 or 1. |
| $\text{select}_b(i)$ | returns the position of the $i$-th $b$ bit in the bitmap where $b$ is 0 or 1. |

By default, $b$ is 1, i.e., $\text{rank}(m) = \text{rank}_1(m)$. These operations are inverses of each other. In other words, $\text{rank}(\text{select}(m)) = \text{select}(\text{rank}(m)) = m$. Since a linear scan is required to process the rank and select functions, the worst-case time complexity will be $\mathcal{O}(n)$.

To clarify how these functions work, consider the binary trees depicted in Figure 3 where the one on the left shows the values and the one on the right shows the numbering of the same tree. If the node has two children, it will be set to 1. Otherwise, it is set to 0. The values of this tree are put in a bit string shown in Figure 4. Figure 5 shows how the position of the left child, right child or parent of a certain node $m$ is computed with the rank and select functions. An example follows:

To find the left child of node 8, we first need to compute rank(8), which is the total number of 1's from node 1 up to and including node 8 and the answer is 7. Therefore, the left child is located in 2*rank(8) = 2*7 = 14 and the right child is in 2*rank(8)+1 = 2*7+1 = 15. The parent of node 8 can be found by computing select($\lfloor 8/2 \rfloor$) or select($\lfloor 4 \rfloor$). The answer can be arrived at by counting the total number of bits starting from node 1, skipping the ones with '0' bits. When we get to node 4 which gives us a total bit count of 4, we then know that node 4 is where the parent of node 8 is.

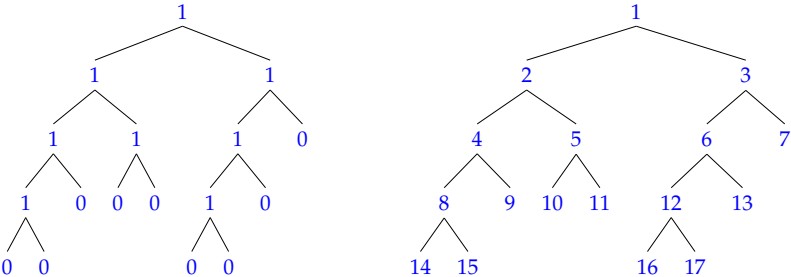

**Figure 3.** A binary tree example for LOUDs. The one on the left shows the values of the nodes and the one on the right shows the same tree with the numbering of the nodes in a left-to-right order. In this case the numbering starts with 1 at the root.

| *m* | 1 | 2 | 3 | 4 | 5 | 6 | 7 | 8 | 9 | 10 | 11 | 12 | 13 | 14 | 15 | 16 | 17 |
|-----|---|---|---|---|---|---|---|---|---|----|----|----|----|----|----|----|----|
| bit | 1 | 1 | 1 | 1 | 1 | 1 | 0 | 1 | 0 | 0 | 0 | 1 | 0 | 0 | 0 | 0 | 0 |

**Figure 4.** A bit string with the values from the binary tree in Figure 3.

| *m* | 1 | 2 | 3 | 4 | 5 | 6 | 8 | 12 |
|-----|---|---|---|---|---|---|---|----|
| Left child($m$) = 2 · rank($m$) | 2 | 4 | 6 | 8 | 10 | 12 | 14 | 16 |
| Right child($m$) = 2 · rank($m$)+1 | 3 | 5 | 7 | 9 | 11 | 13 | 15 | 17 |
| Parent(m) = select($\lfloor m/2 \rfloor$) | - | 1 | 1 | 2 | 2 | 3 | 4 | 6 |

**Figure 5.** With the rank and select functions listed in the first column, we can navigate the binary tree in Figure 3 and compute the position node for the left child, right child or parent of the node.

In the next section, we will explain how the rank function can be used to determine the children's positions in a $k^2$-tree, thus enabling us to query the values of the cells.

### 2.3. $k^2$-Tree

Originally proposed for compressing Web graphs, $k^2$-tree is a LOUDS variant compact data structure [22]. The tree represents a binary adjacency matrix of a graph (see Figure 2). It is constructed by recursively partitioning the matrix into square submatrices of equal size until each submatrix reaches a size of $k$ x $k$ where $k \geq 2$. During the process of partitioning, if there is at least one cell in the submatrix that has a value of 1, the node in the tree will be set to 1. Otherwise, it will be set to 0 (i.e., it is a leaf and has no children) and this particular submatrix will not be partitioned any further. Figure 2 illustrates an example of a graph of 6 nodes, its 8 × 8 binary adjacency matrix at various stages of recursive partitioning, and the $k^2$-tree that is constructed from the matrix.

The values of $k^2$-trees are basically stored in two bitmaps denoted by $T$ (tree) and $L$ (leaves). The values are traversed in a breadth-first fashion starting with the first level. The $T$ bitmap stores the bits at all levels except the last one where its bits will be stored in the $L$ bitmap. Note that the bit values of $T$ which are either 0 or 1 will be stored as a bit vector. To illustrate this with an example, we again make use of the binary matrix in Figure 2. The $T$ bitmap contains all the bits from levels 1 and 2. Thus the $T$ bitmap has the following bits: 1001 1101 1000 (see Figure 6). The bits from the last level, level 3, will be stored in the $L$ bitmap with the following bits: 1001 0001 1011 1111.

Consider a set $S$ with elements from 1 to $n$, to find the child's or the parent's position of a certain node $m$ in a $k^2$-tree, we perform the following operations:

first-child$(m) \leftarrow$ rank$(m) \cdot k^2$ where $1 \leq m \leq \|S\|$

parent$(m) \leftarrow$ select$(\lfloor m/k^2 \rfloor)$ where $1 \leq m \leq \|S\|$

Once again using the $k^2$-tree in Figure 2 as an example, with the $T$ bitmap (Figure 6) and the rank and select functions, we can navigate the tree and obtain the positions of the first child and the parent. Figure 7 shows how the nodes of the $k^2$-tree are numbered.

Ex.    Locate the first child of node 8:

rank$_1(8) * 4 = 6 * 4 = 24$

(There are 6 one bits in the $T$ bitmap starting from node 0 up to and including node 8.)

Ex.    Locate the parent of node 11:

select$_1(\lfloor 11/4 \rfloor) =$ select$_1(2) = 3$

(Start counting from node 0, skipping all nodes with '0' bits, and node 3 is the first node that gives a total number of 1-bit count of 2. Therefore, node 3 is the parent.

| Node | 0 | 1 | 2 | 3 | 4 | 5 | 6 | 7 | 8 | 9 | 10 | 11 |
|------|---|---|---|---|---|---|---|---|---|---|----|----|
| Bit  | 1 | 0 | 0 | 1 | 1 | 1 | 0 | 1 | 1 | 0 | 0  | 0  |

**Figure 6.** A $T$ bitmap with the first node labeled as 0.

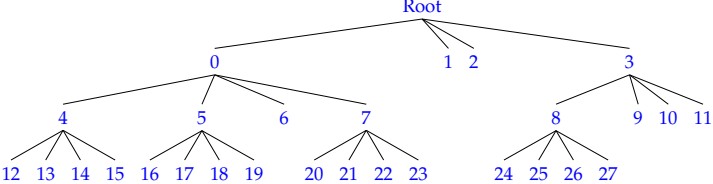

**Figure 7.** An example showing how the rank function is computed to obtain the children's position on a $k^2$-tree node (k=2) based on the tree in Figure 2. It starts with 0 on the first child of the root (first level) and the numbering traverses from left to right and from top to bottom.

It was shown that $k^2$-tree gave the best performance when the matrix was sparse with large clusters of 0's or 1's [20].

### 2.4. DACs

This section describes DACs which is used in $k^2$-raster to directly access variable-length codes. Based on the concept of compact data structures, DACs were proposed in the papers published by Brisaboa et al. in 2009 and 2013 [23,24] and the structure was proven to yield good compression ratios for variable-length integer sequences. By means of the rank function, it gains fast direct access to any position of the sequence in a very compact space. The original authors also asserted that it was better suited for a sequence of integers with a skewed frequency distribution toward smaller integer values.

Different types of encoding are used for DACs and the one that we are interested in for $k^2$-raster is called Vbyte coding. Consider a sequence of integers $x$. Each integer, which is represented by $\lfloor \log_2 x_i \rfloor + 1$ bits, is broken into blocks of bits of size $S$. Each block is stored as a chunk of $S + 1$ bits. The chunk that holds the most significant bits has the highest bit set to 0 while the other chunks have their highest bit set to 1. For example, if we have an integer 20 ($10100_2$) which is 5 bits long and if the block size is $S = 3$, then we can have 2 chunks denoted by the following: $\underline{0}010 \ \underline{1}100$.

To show how the chunks are organized and stored, we again illustrate it with an example. If we have 3 integers of variable length 20 ($10100_2$), 6 ($110_2$), 73 ($1001001_2$) and each block size is 3, then the three integers have the following representations.

20    $\underline{0}010 \ \underline{1}100$ (B$_{1,2}$A$_{1,2}$ B$_{1,1}$A$_{1,1}$)

6    $\underline{0}110$ (B$_{2,1}$A$_{2,1}$)

73    $\underline{0}001 \ \underline{1}001 \ \underline{1}001$ (B$_{3,3}$A$_{3,3}$ B$_{3,2}$A$_{3,2}$ B$_{3,1}$A$_{3,1}$)

We will store them in three chunks of arrays *A* and bitmaps *B*. This is depicted in Figure 8. To retrieve the values in the arrays *A*, we make use of the corresponding bitmaps *B* with the rank function.

More information on DACs and the software code can be found in the papers [23,24].

| $C_1$ | $A_1$ | 100 ($A_{1,1}$) | 110 ($A_{2,1}$) | 001 ($A_{3,1}$) |
|---|---|---|---|---|
| | $B_1$ | 1 ($B_{1,1}$) | 0 ($B_{2,1}$) | 1 ($B_{3,1}$) |
| $C_2$ | $A_2$ | 010 ($A_{1,2}$) | 001 ($A_{3,2}$) | |
| | $B_2$ | 0 ($B_{1,2}$) | 1 ($B_{3,2}$) | |
| $C_3$ | $A_3$ | 001 ($A_{3,3}$) | | |
| | $B_3$ | 0 ($B_{3,3}$) | | |

**Figure 8.** Organization of 3 Directly Addressable Codes (DACs) clusters.

## 2.5. $k^2$-Raster

$k^2$-raster is a compact data structure that allows us to store raster pixels in reduced space. It consists of several basic components: bitmaps, DACs and LOUDS. Similar to a $k^2$-tree, the image matrix is partitioned recursively until each subquadrant is of size $k^2$. The resulting LOUDS tree topology contains the bitmap *T* where the elements are accessed with the rank function. Unlike $k^2$-tree, at each tree level, the maximum and minimum values of each subquadrant are stored in two bitmaps which are respectively called *Vmax* and *Vmin*. However, to compress the structure further, the maximum and minimum values of each level are compared with the corresponding values of the parent and their differences will replace the stored values in the *Vmax* and *Vmin* bitmaps. The rationale behind all this is to obtain smaller values for each node so as to get a better compression with DACs. An example of a simple 8 × 8 matrix is given to illustrate this point in Figure 9. A $k^2$-raster is constructed from this matrix with maximum and minimum values stored in each node in Figure 10. The structure is further modified, according to the above discussion, to form a tree with smaller maximum and minimum values and this is shown in Figure 11.

Next, with the exception of the root node at the top level, the *Vmax* and *Vmin* bitmaps at all levels are concatenated to form *Lmax* and *Lmin* bitmaps. The root's maximum (*rMax*) and minimum (*rMin*) values are integer values and will remain uncompressed.

For an image of size *n* × *n* with *n* bands, the time complexity to build all the $k^2$-rasters is $\mathcal{O}(n^3)$ [22]. To query a cell from the structure, which has a tree height of at most $\lceil \log_k n \rceil$ levels, the time complexity to extract a codeword at a single *Lmax* level is $\mathcal{O}(\log_k n)$, and this is the worst-case time to traverse from the root node to the last level of the structure. The number of levels, $\mathcal{L}$, in *Lmax* can be obtained from the maximum integer in the sequence and with this, we can compute the time complexity for a cell query, which is $\mathcal{O}(\log_k n \cdot \mathcal{L})$ [23,25].

To sum up, a $k^2$-raster structure is composed of a bitmap *T*, a maximum bitmap *Lmax*, a minimum bitmap *Lmin*, a root maximum *rMax* integer value and a root minimum *rMin* integer value.

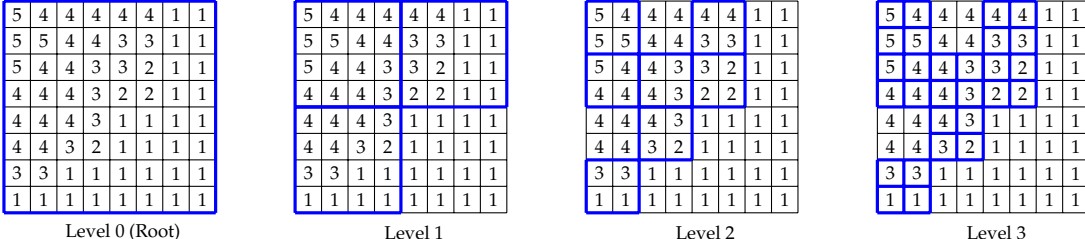

**Figure 9.** An example of an $8 \times 8$ matrix for $k^2$-raster. The matrix is recursively partitioned into square subquadrants of equal size. During the process, unless all the cells in a subquadrant have the same value, the partitioning will continue. Otherwise the partitioning of this particular subquadrant will end at this point.

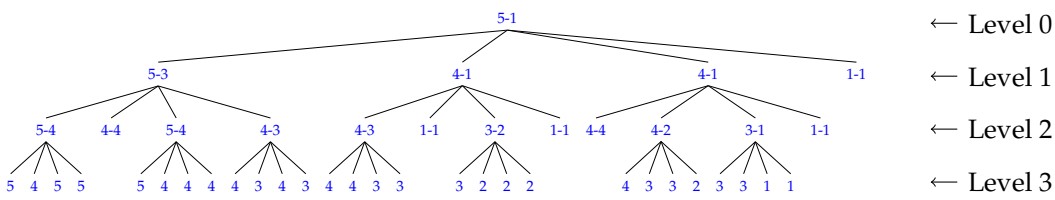

**Figure 10.** A $k^2$-raster ($k = 2$) tree storing the maximum and mininum values for each quadrant of every recursive subdivision of the matrix in Figure 9. Every node contains the maximum and minimum values of the subquadrant, separated by a dash. On the last level, only one value is shown as each subquadrant contains only one cell.

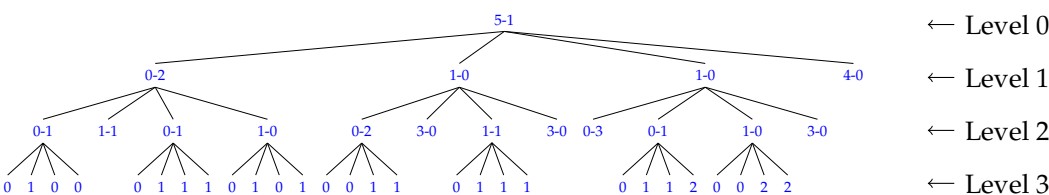

**Figure 11.** Based on the tree in Figure 10, the maximum value of each node is subtracted from that of its parent while the minimum value of the parent is subtracted from the node's minimum value. These differences will replace their corresponding values in the node. The maximum and minimum values of the root remain the same.

## 2.6. Predictive Method

As mentioned in the Introduction, an interband predictor called 3D-CALIC was proposed by Wu et al. in 2000 and another predictor called M-CALIC by Magli et al. in 2004. Our predictor is based on the idea of least squares method and the use of reference bands that were discussed in both the 3D-CALIC [14] and M-CALIC [15] papers. Consider two neighboring or close neighboring bands of the same hyperspectral image. These bands can be represented by two vectors $\mathbf{X} = (x_1, x_2, x_3, ..., x_{n-1}, x_n)$ and $\mathbf{Y} = (y_1, y_2, y_3, ..., y_{n-1}, y_n)$ where $x_i$ and $y_i$ are two pixels that are located at the same spatial position but in different bands, and $n$ is the number of pixels in each band. We can then employ the close similarity between the bands to predict the pixel value in the current band $\mathbf{Y}$ using the corresponding pixel value in band $\mathbf{X}$, which we designate as the reference band.

A predictor for a particular band can be built from the linear equation:

$$\hat{\mathbf{Y}} = \alpha \mathbf{X} + \beta \tag{1}$$

so as to minimize $||\hat{\mathbf{Y}} - \mathbf{Y}||_2^2$ where $\hat{\mathbf{Y}}$ is the predicted value and $\mathbf{Y}$ is the actual value of the current band. The optimal values for $\alpha$ and $\beta$ should minimize the prediction error of the current pixel and can be obtained by using the least squares solution:

$$\hat{\alpha} = \frac{n \sum_{i=1}^{n} x_i y_i - \sum_{i=1}^{n} x_i \sum_{i=1}^{n} y_i}{n \sum_{i=1}^{n} x_i^2 - (\sum_{i=1}^{n} x_i)^2} \, , \tag{2}$$

$$\hat{\beta} = \frac{n \sum_{i=1}^{n} y_i - \hat{\alpha} \sum_{i=1}^{n} x_i}{n} \tag{3}$$

where $n$ is the size of each band, i.e., the height multiplied by the width, $\hat{\alpha}$ the optimal value of $\alpha$ and $\hat{\beta}$ the optimal value of $\beta$.

The difference between the actual and predicted pixel values of a band is known as the residual value or the prediction error. When all the pixel values in the current band are calculated, these prediction residuals will be saved in a vector, which will later be used as input to a $k^2$-raster.

In other words, for a particular pixel in the current band and the corresponding pixel in the reference band, $\delta_i$ being the residual value, $y_i$ the actual value of the current band, and $x_i$ the value of the reference band, to encode, the following equation is used:

$$\delta_i = y_i - (\hat{\alpha} \cdot x_i + \hat{\beta}) \, . \tag{4}$$

To decode, the following equation is used:

$$y_i = \delta_i + (\hat{\alpha} \cdot x_i + \hat{\beta}) \, . \tag{5}$$

The distance from the reference band affects the residual values. The closer the current band is to the reference band, the smaller the residual values would tend to be. We can arrange the bands into groups. For example, the first band can be chosen as the reference and the second, third and fourth bands will have their residual values calculated with respect to the first band. And the next group starts with the fifth band as the reference band, etc.

For this coding method, the group size (stored as a 2-byte short integer) as well as the $\hat{\alpha}$ and $\hat{\beta}$ values for each band (stored as 8-byte double's) will need to be saved for use in both the encoder and the decoder. Note that the size of these extra data is insignificant - which generally comes to around 3.5 kB - compared to the overall size of the structure.

*2.7. Differential Method*

In the differential encoding, which is a special case of the predictor where $\alpha = 1$ and $\beta = 0$, the residual value is obtained by simply taking the difference between the reference band and the current band. For a particular pixel in the current band and the corresponding pixel in the reference band, $\delta_i$ being the residual value, $y_i$ the actual value of the current band, and $x_i$ the value of the reference band, to encode, the following equation is used:

$$\delta_i = y_i - x_i \, . \tag{6}$$

To decode, the following equation is used:

$$y_i = \delta_i + x_i \, . \tag{7}$$

Like the predictor, we can use the first band as the reference band and the next several bands can use this reference band to find the residual values. Again, the grouping is repeated up to the last band. For this coding method, only the group size (stored as a 2-byte short integer) needs to be saved.

*2.8. Related Work*

Since the publication of the proposals on $k^2$-tree and $k^2$-raster, more research has been done to extend the capabilities of the structures to 3D where the first and second dimensions represent the spatial element and the third dimension the time element.

Based on their previous research of $k^2$-raster, Silva-Coira et al. [26] proposed a structure called Temporal $k^2$-raster (T $- k^2$raster) which represents a time-series of rasters. It takes advantage of the fact that in a time-series, the values in a matrix $M_1$ are very close to, if not the same as, the next matrix $M_2$ or even the one after that, $M_3$, along the timeline. The matrices can then be grouped into $\tau$ time instants where the values of the elements of the first matrix in the group is subtracted from the corresponding ones in the current matrix. The result will be smaller integer values that would help form a more compact tree as there are likely to be more zeros in the tree than before. Their experimental results bear this out. When the $\tau$ value is small ($\tau = 4$), the sizes are small. However, as would be expected, the results are not as favorable when the $\tau$ value becomes larger ($\tau = 50$). Akin to the Temporal $k^2$-raster, the differential encoding on $k^2$-raster that we are proposing in this paper also exploits the similarity between neighboring matrices or bands in a hyperspectral image to form a more compact structure.

Another study on compact representation of raster images in a time-series was proposed earlier this year by Cruces et al. in [27]. This method is based on the 3D to 2D mapping of a raster where 3D tuples $<x, y, z>$ are mapped into a 2D binary grid. That is, a raster of size $w \times h$ with values in a certain range, between 0 and $v$ inclusive will have a binary matrix of $w \times h$ columns and $v+1$ rows. All the rasters will then be concatenated into a 3D cube and stored as a $k^3$-tree.

## 3. Results

In this section we describe some of the experiments that were performed to show the use of compact data structures, prediction and differential encoding for real-time processing and compression. First, we show the results with other compression algorithms and techniques that are currently in use such as gzip, bzip2, xz, M-CALIC [15] and CCSDS 123.0-B-1 [9]. Then we compare the build time and the data access time for $k^2$-raster with and without prediction and differential encoding. Next, we show the results of different rates in $k^2$-raster that are produced as different k-values are applied. Similarly, the results of different group sizes for prediction and differential encoding are shown. Finally, the predictive method and the differential method are compared.

Experiments were conducted using hyperspectral images from different sensors: Atmospheric Infrared Sounder (AIRS), Airborne Visible/Infrared Imaging Spectrometer (AVIRIS), Compact Reconnaissance Imaging Spectrometer for Mars (CRISM), Hyperion, and Infrared Atmospheric Sounding Interferometer (IASI). Except for IASI, all of them are publicly available for download (http://cwe.ccsds.org/sls/docs/sls-dc/123.0-B-Info/TestData). Table 1 gives more detailed information on these images. The table also shows the bit-rate reduction for using $k^2$-raster with and without prediction. Performance in terms of bit rate and entropy is evaluated for them.

For best results in $k^2$-raster for the testing images, we used the optimal $k$-value, and also in the case of the predictor and the differential encoder, the optimal group size for each image was used. The effects of using different $k$-values and different group sizes will be discussed and tested in two of the subsections below.

To build the structure of $k^2$-raster and the cell query functions, a program in C was written. The algorithms presented in the paper by Ladra et al. [20] were the basis and reference for writing the code. The DACs software that was used in conjunction with our program is available at the Universidade da Coruña's Database Laboratory (Laboratorio de Bases de Datos) website (http://lbd.udc.es/research/DACS/). The package is called "DACs, optimization with no further restrictions". As for the predictive and differential methods, another C program was written to perform the tasks needed to give us the results that we will discuss below. All the code was compiled using gcc or g++ 5.4.0 20160609 with -Ofast optimization.

**Table 1.** Hyperspectral images used in our experiments. It also shows the bit rate and bit rate reduction using $k^2$-raster with and without the predictor. $x$ is the image width, $y$ the image height and $z$ the number of spectral bands. The unit bpppb stands for bits per pixel per band.

| Sensor | Name | C/U* | Acronym | Original Dimensions ($x \times y \times z$) | Bit Depth (bpppb) | Optimal $k$-Value | $k^2$-Raster Bit Rate (bpppb) | $k^2$-Raster Bit-Rate Reduction (%) | $k^2$-Raster+ Predictor Bit Rate (bpppb) | $k^2$-Raster+ Predictor Bit-Rate Reduction (%) |
|---|---|---|---|---|---|---|---|---|---|---|
| AIRS | 9 | U | AG9 | 90 × 135 × 1501 | 12 | 6 | 9.49 | 21% | 6.76 | 44% |
| | 16 | U | AG16 | 90 × 135 × 1501 | 12 | 6 | 9.12 | 24% | 6.63 | 45% |
| | 60 | U | AG60 | 90 × 135 × 1501 | 12 | 6 | 9.81 | 18% | 7.06 | 41% |
| | 126 | U | AG126 | 90 × 135 × 1501 | 12 | 6 | 9.61 | 20% | 7.05 | 41% |
| | 129 | U | AG129 | 90 × 135 × 1501 | 12 | 6 | 8.65 | 28% | 6.47 | 46% |
| | 151 | U | AG151 | 90 × 135 × 1501 | 12 | 6 | 9.53 | 21% | 7.02 | 41% |
| | 182 | U | AG182 | 90 × 135 × 1501 | 12 | 6 | 9.68 | 19% | 7.19 | 40% |
| | 193 | U | AG193 | 90 × 135 × 1501 | 12 | 6 | 9.44 | 21% | 7.06 | 41% |
| AVIRIS | Yellowstone sc. 00 | C | ACY00 | 677 × 512 × 224 | 16 | 6 | 9.61 | 40% | 6.87 | 57% |
| | Yellowstone sc. 03 | C | ACY03 | 677 × 512 × 224 | 16 | 6 | 9.42 | 41% | 6.72 | 58% |
| | Yellowstone sc. 10 | C | ACY10 | 677 × 512 × 224 | 16 | 4 | 7.57 | 53% | 5.84 | 64% |
| | Yellowstone sc. 11 | C | ACY11 | 677 × 512 × 224 | 16 | 6 | 8.81 | 45% | 6.52 | 59% |
| | Yellowstone sc. 18 | C | ACY18 | 677 × 512 × 224 | 16 | 6 | 9.78 | 39% | 7.04 | 56% |
| | Yellowstone sc. 00 | U | AUY00 | 680 × 512 × 224 | 16 | 9 | 11.92 | 25% | 9.04 | 44% |
| | Yellowstone sc. 03 | U | AUY03 | 680 × 512 × 224 | 16 | 9 | 11.74 | 27% | 8.87 | 45% |
| | Yellowstone sc. 10 | U | AUY10 | 680 × 512 × 224 | 16 | 9 | 9.99 | 38% | 8.00 | 50% |
| | Yellowstone sc. 11 | U | AUY11 | 680 × 512 × 224 | 16 | 9 | 11.27 | 30% | 8.77 | 45% |
| | Yellowstone sc. 18 | U | AUY18 | 680 × 512 × 224 | 16 | 9 | 12.15 | 24% | 9.29 | 42% |
| CRISM | frt000065e6_07_sc164 | U | C164 | 640 × 420 × 545 | 12 | 6 | 10.08 | 16% | 10.02 | 16% |
| | frt00008849_07_sc165 | U | C165 | 640 × 450 × 545 | 12 | 6 | 10.37 | 14% | 10.33 | 14% |
| | frt0001077d_07_sc166 | U | C166 | 640 × 480 × 545 | 12 | 6 | 11.05 | 8% | 11.08 | 8% |
| | hrl00004f38_07_sc181 | U | C181 | 320 × 420 × 545 | 12 | 5 | 9.97 | 17% | 9.52 | 21% |
| | hrl0000648f_07_sc182 | U | C182 | 320 × 450 × 545 | 12 | 5 | 10.11 | 16% | 9.84 | 18% |
| | hrl0000ba9c_07_sc183 | U | C183 | 320 × 480 × 545 | 12 | 5 | 10.65 | 11% | 10.59 | 12% |

**Table 1.** *Cont.*

| Sensor | Name | C/U[★] | Acronym | Original Dimensions ($x \times y \times z$) | Bit Depth (bpppb) | Optimal $k$-Value | $k^2$-raster Bit Rate (bpppb) | $k^2$-Raster Bit-Rate Reduction (%) | $k^2$-Raster+ Predictor Bit Rate (bpppb) | $k^2$-Raster+ Predictor Bit-Rate Reduction (%) |
|---|---|---|---|---|---|---|---|---|---|---|
| Hyperion | Agricultural 2905 [†] | C | HCA1 | $256 \times 2905 \times 242$ | 12 | 8 | 8.20 | 32% | 7.47 | 38% |
| | Agricultural 3129 [†] | C | HCA2 | $256 \times 3129 \times 242$ | 12 | 8 | 8.08 | 33% | 7.50 | 37% |
| | Coral Reef [†] | C | HCC | $256 \times 3127 \times 242$ | 12 | 8 | 7.38 | 39% | 7.41 | 38% |
| | Urban [†] | C | HCU | $256 \times 2905 \times 242$ | 12 | 8 | 8.59 | 28% | 7.83 | 35% |
| | Filtered Erta Ale [†] | U | HFUEA | $256 \times 3187 \times 242$ | 12 | 8 | 6.84 | 43% | 5.99 | 50% |
| | Filtered Lake Monona [†] | U | HFULM | $256 \times 3176 \times 242$ | 12 | 8 | 6.79 | 43% | 6.06 | 49% |
| | Filtered Mt. St. Helena [†] | U | HFUMS | $256 \times 3242 \times 242$ | 12 | 8 | 6.78 | 43% | 5.88 | 51% |
| | Erta Ale [†] | U | HUEA | $256 \times 3187 \times 242$ | 12 | 8 | 7.57 | 37% | 6.99 | 42% |
| | Lake Monona [†] | U | HULM | $256 \times 3176 \times 242$ | 12 | 8 | 7.52 | 37% | 7.08 | 41% |
| | Mt. St. Helena [†] | U | HUMS | $256 \times 3242 \times 242$ | 12 | 8 | 7.49 | 38% | 6.93 | 42% |
| IASI | Level 0 1 [‡] | U | I01 | $60 \times 1528 \times 8359$ | 12 | 4 | 5.93 | 51% | 4.69 | 61% |
| | Level 0 2 [‡] | U | I02 | $60 \times 1528 \times 8359$ | 12 | 4 | 5.90 | 51% | 4.75 | 60% |
| | Level 0 3 [‡] | U | I03 | $60 \times 1528 \times 8359$ | 12 | 4 | 5.42 | 55% | 4.58 | 62% |
| | Level 0 4 [‡] | U | I04 | $60 \times 1528 \times 8359$ | 12 | 4 | 6.23 | 48% | 4.90 | 59% |

★: Calibrated or Uncalibrated; †: Cropped to $256 \times 512 \times 242$; ‡: Cropped to $60 \times 256 \times 8359$.

The machine that these experiments ran on has an Intel Core 2 Duo CPU E7400 @2.80GHz with 3072KB of cache and 3GB of RAM. The operating system is Ubuntu 16.04.5 LTS with kernel 4.15.0-47-generic (64 bits).

To ensure that there was no loss of information, the image was reconstructed by reverse transformation and verified to be identical to the original image in the case of predictive and differential methods. For $k^2$-raster, after saving the structure to disk, we made sure that the original image could be reconstructed from the saved data.

### 3.1. Comparison with Other Compression Algorithms

Both $k^2$-raster with and without predictive and differential encoding were compared to other commonly-used compression algorithms such as gzip, bzip2, xz, and specialized algorithms such as M-CALIC and CCSDS 123.0-B-1. The results for the comparison are shown in Table 2 and depicted in Figure 12.

It can be seen that $k^2$-raster alone already performed better than gzip. When it was used with the predictor, it produced a bit rate that was basically on a par with and sometimes better than other compression algorithms such as xz or bzip2. However, it could not attain the bit-rate level done by CCSDS 123.0-B-1 or M-CALIC. This was to be expected as both are specialized compression techniques, and CCSDS 123.0-B-1 is considered a baseline against which all compression algorithms for hyperspectral images are measured. Nevertheless, $k^2$-raster provides direct access to the elements without full decompression, and this is undoubtedly the major advantage it has over all the aforementioned compression algorithms.

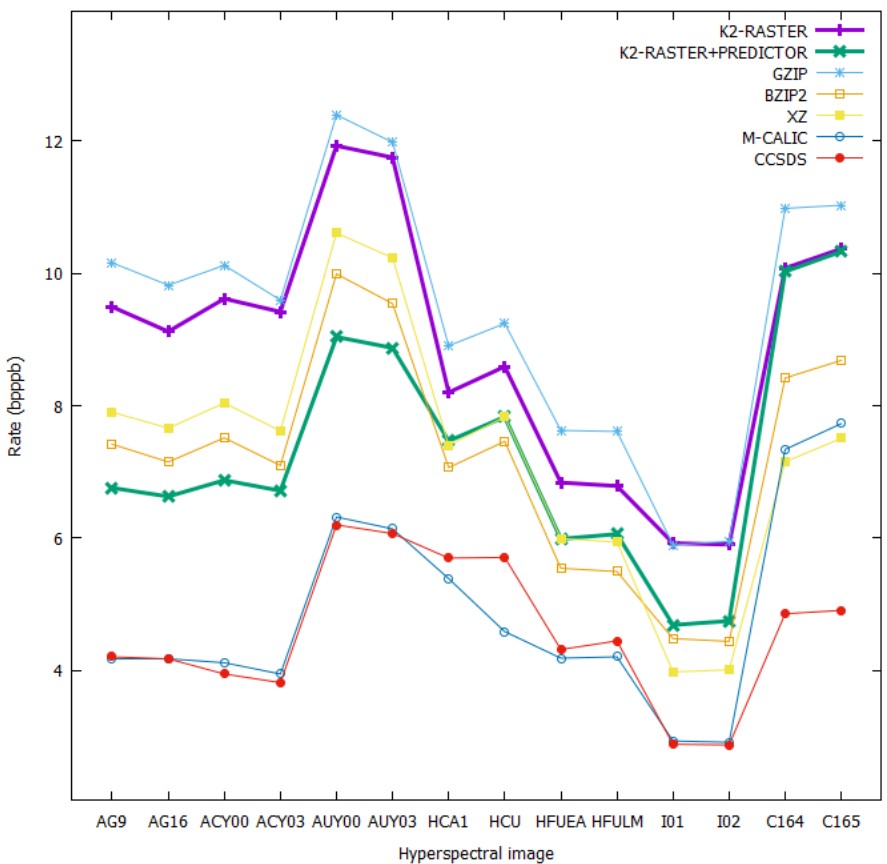

**Figure 12.** A rate (bpppb) comparison with other compression techniques.

**Table 2.** A rate (bpppb) comparison with other compression techniques. The optimal values for all compression algorithms (except for M-CALIC, CCSDS 123.0-B-1) are highlighted in red. Results for CCSDS 123.0-B-1 are from [28].

| Sensor | Name | C/U ★ | Acronym | Compression Technique (bpppb) | | | | | | | |
| | | | | $k^2$-Raster | $k^2$-Raster + Predictor | $k^2$-Raster + Differential | gzip | bzip2 | xz | M-CALIC | CCSDS 123.0-B-1 |
|---|---|---|---|---|---|---|---|---|---|---|---|
| AIRS | 9 | U | AG9 | 9.49 | <span style="color:red">6.76</span> | 7.52 | 10.16 | 7.42 | 7.90 | 4.19 | 4.21 |
| | 16 | U | AG16 | 9.12 | <span style="color:red">6.63</span> | 7.29 | 9.82 | 7.15 | 7.66 | 4.19 | 4.18 |
| | 60 | U | AG60 | 9.81 | <span style="color:red">7.06</span> | 7.82 | 10.53 | 7.71 | 8.23 | 4.41 | 4.36 |
| | 126 | U | AG126 | 9.61 | <span style="color:red">7.05</span> | 7.78 | 10.33 | 7.64 | 8.10 | 4.39 | 4.38 |
| | 129 | U | AG129 | 8.65 | <span style="color:red">6.47</span> | 6.96 | 9.50 | 6.68 | 7.22 | 4.08 | 4.12 |
| | 151 | U | AG151 | 9.53 | <span style="color:red">7.02</span> | 7.74 | 10.31 | 7.43 | 7.97 | 4.39 | 4.41 |
| | 182 | U | AG182 | 9.68 | <span style="color:red">7.19</span> | 7.94 | 10.64 | 7.79 | 8.33 | 4.45 | 4.42 |
| | 193 | U | AG193 | 9.44 | <span style="color:red">7.06</span> | 7.77 | 10.15 | 7.47 | 7.94 | 4.42 | 4.42 |
| AVIRIS | Yellowstone sc. 00 | C | ACY00 | 9.61 | <span style="color:red">6.87</span> | 7.79 | 10.12 | 7.51 | 8.04 | 4.12 | 3.95 |
| | Yellowstone sc. 03 | C | ACY03 | 9.42 | <span style="color:red">6.72</span> | 7.65 | 9.59 | 7.10 | 7.62 | 3.95 | 3.82 |
| | Yellowstone sc. 10 | C | ACY10 | 7.57 | 5.84 | 6.26 | 7.41 | <span style="color:red">5.30</span> | 5.73 | 3.31 | 3.36 |
| | Yellowstone sc. 11 | C | ACY11 | 8.81 | <span style="color:red">6.52</span> | 6.85 | 9.04 | 6.65 | 7.07 | 3.71 | 3.63 |
| | Yellowstone sc. 18 | C | ACY18 | 9.78 | <span style="color:red">7.04</span> | 7.53 | 10.00 | 7.45 | 7.95 | 4.09 | 3.90 |
| | Yellowstone sc. 00 | U | AUY00 | 11.92 | <span style="color:red">9.04</span> | 10.04 | 12.39 | 9.99 | 10.61 | 6.32 | 6.20 |
| | Yellowstone sc. 03 | U | AUY03 | 11.74 | <span style="color:red">8.87</span> | 9.91 | 11.98 | 9.54 | 10.23 | 6.14 | 6.07 |
| | Yellowstone sc. 10 | U | AUY10 | 9.99 | 8.00 | 8.57 | 10.17 | <span style="color:red">7.71</span> | 8.40 | 5.53 | 5.58 |
| | Yellowstone sc. 11 | U | AUY11 | 11.27 | <span style="color:red">8.77</span> | 9.21 | 11.49 | 9.08 | 9.66 | 5.91 | 5.84 |
| | Yellowstone sc. 18 | U | AUY18 | 12.15 | <span style="color:red">9.29</span> | 9.92 | 12.29 | 9.90 | 10.58 | 6.33 | 6.21 |
| CRISM | frt000065e6_07_sc164 | U | C164 | 10.08 | 10.02 | 10.06 | 10.98 | 8.42 | <span style="color:red">7.15</span> | 7.34 | 4.86 |
| | frt00008849_07_sc165 | U | C165 | 10.37 | 10.33 | 10.37 | 11.03 | 8.68 | <span style="color:red">7.51</span> | 7.73 | 4.91 |
| | frt0001077d_07_sc166 | U | C166 | 11.05 | 11.08 | 11.14 | 11.20 | 9.04 | <span style="color:red">7.64</span> | 8.44 | 5.44 |
| | hrl00004f38_07_sc181 | U | C181 | 9.97 | 9.52 | 9.52 | 10.77 | 8.28 | <span style="color:red">8.20</span> | 7.09 | 4.27 |
| | hrl0000648f_07_sc182 | U | C182 | 10.11 | 9.84 | 9.86 | 10.90 | 8.53 | <span style="color:red">7.90</span> | 7.28 | 4.49 |
| | hrl0000ba9c_07_sc183 | U | C183 | 10.65 | 10.59 | 10.64 | 10.87 | 8.52 | <span style="color:red">7.28</span> | 7.91 | 4.96 |

**Table 2.** *Cont.*

| Sensor | Name | C/U [*] | Acronym | Compression Technique (bpppb) | | | | | | | |
|--------|------|---------|---------|-----------|-----------|-----------|------|-------|------|---------|-------------|
| | | | | $k^2$-Raster | $k^2$-Raster + Predictor | $k^2$-Raster + Differential | gzip | bzip2 | xz | M-CALIC | CCSDS 123.0-B-1 |
| Hyperion | Agricultural 2905 [†] | C | HCA1 | 8.20 | 7.47 | 7.47 | 8.90 | 7.07 | 7.40 | 5.39 | - |
| | Agricultural 3129 [†] | C | HCA2 | 8.08 | 7.50 | 7.50 | 8.84 | 7.04 | 7.35 | 5.28 | 5.70 |
| | Coral Reef [†] | C | HCC | 7.38 | 7.41 | 7.41 | 7.45 | 5.74 | 5.90 | 4.59 | 5.42 |
| | Urban [†] | C | HCU | 8.59 | 7.83 | 7.83 | 9.24 | 7.46 | 7.83 | 5.25 | 5.71 |
| | Filtered Erta Ale [†] | U | HFUEA | 6.84 | 5.99 | 6.15 | 7.63 | 5.55 | 6.00 | 4.19 | 4.32 |
| | Filtered Lake Monona [†] | U | HFULM | 6.79 | 6.06 | 6.18 | 7.61 | 5.50 | 5.94 | 4.21 | 4.45 |
| | Filtered Mt. St. Helena [†] | U | HFUMS | 6.78 | 5.88 | 6.15 | 7.18 | 5.44 | 5.74 | 4.11 | 4.35 |
| | Erta Ale [†] | U | HUEA | 7.57 | 6.99 | 7.06 | 8.69 | 6.41 | 6.73 | 4.87 | 4.32 |
| | Lake Monona [†] | U | HULM | 7.52 | 7.08 | 7.13 | 8.69 | 6.46 | 6.74 | 4.94 | 4.45 |
| | Mt. St. Helena [†] | U | HUMS | 7.49 | 6.93 | 7.04 | 8.26 | 6.28 | 6.48 | 4.82 | 4.36 |
| IASI | Level 0 1 [‡] | U | I01 | 5.93 | 4.69 | 5.01 | 5.90 | 4.48 | 3.98 | 2.94 | 2.89 |
| | Level 0 2 [‡] | U | I02 | 5.90 | 4.75 | 5.03 | 5.96 | 4.44 | 4.01 | 2.92 | 2.88 |
| | Level 0 3 [‡] | U | I03 | 5.42 | 4.58 | 4.79 | 5.25 | 3.94 | 3.75 | 2.92 | 2.88 |
| | Level 0 4 [‡] | U | I04 | 6.23 | 4.90 | 5.20 | 6.30 | 4.71 | 4.24 | 2.97 | 2.90 |

[*]: Calibrated or Uncalibrated; [†]: Cropped to 256 × 512 × 242 except for CCSDS 123.0; [‡]: Cropped to 60 × 256 × 8359 except for CCSDS 123.0.

*3.2. Build Time*

Both the time to build the $k^2$-raster only and the time to build $k^2$-raster with predictive and differential preprocessing were measured. They were then compared against the time to compress the data with gzip. The results are presented in Table 3. We can see that the build time for $k^2$-raster only took half as long as with gzip. Comparing the predictive and the differential methods, the time difference is small although it generally took longer to build the former than the latter due to the additional time needed to compute the values of $\hat{\alpha}$ and $\hat{\beta}$. Both, however, still took less time to build than gzip compression.

**Table 3.** A comparison of build time (in seconds) using $k^2$-raster only and $k^2$-raster with predictive and differential methods.

| Hyperspectral Image | Build Time (s) | | | Gzip Compression (s) |
|---|---|---|---|---|
| | $k^2$-Raster | $k^2$-Raster + Predictor | $k^2$-Raster + Differential | |
| AG9 | 1.86 | 2.23 | 2.12 | 3.18 |
| AG16 | 1.78 | 2.22 | 2.09 | 3.49 |
| ACY00 | 8.32 | 10.11 | 9.49 | 15.01 |
| ACY03 | 8.26 | 10.00 | 9.47 | 15.32 |
| AUY00 | 5.56 | 7.39 | 6.84 | 12.10 |
| AUY03 | 5.59 | 7.38 | 6.76 | 12.68 |
| C164 | 17.84 | 21.32 | 21.59 | 27.94 |
| C165 | 17.89 | 22.83 | 22.92 | 30.83 |
| HCA1 | 1.98 | 2.67 | 2.47 | 5.59 |
| HCA2 | 1.98 | 2.64 | 2.42 | 5.80 |
| HFUEA | 2.38 | 3.01 | 3.05 | 7.59 |
| HFULM | 2.41 | 3.04 | 2.87 | 7.57 |
| HFUMS | 2.33 | 2.95 | 2.76 | 8.26 |
| I01 | 14.58 | 18.62 | 16.56 | 31.59 |
| I02 | 14.66 | 17.49 | 16.66 | 29.64 |

*3.3. Access Time*

Several tests were conducted to see what the access time was like to query the cells in each image and we found that the time for a random cell access took longer for a predictor compared to just using the $k^2$-raster. This was expected but we should bear in mind that the bit rates are reduced when a predictor is used, thus decreasing storage size and transmission rate. Note that the last column also lists the time to decompress a gzip image file and it took at least 4 or 5 times longer than using a predictor to randomly access the data $10^5$ times. Table 4 shows the results of access time in milliseconds for 100,000 iterations of random cell query done by getCell(), a function which was described in the paper from Ladra et al. [20] for accessing pixel values in a $k^2$-raster.

**Table 4.** A comparison of access time (in milliseconds) using $k^2$-raster only and $k^2$-raster with predictive and differential encoders.

| Hyperspectral Image | 100,000 Iterations of Random Access (ms) | | | Gzip Decompression (ms) |
|---|---|---|---|---|
| | $k^2$-raster | $k^2$-raster + Predictor | $k^2$-raster + Differential | |
| AG9 | 90 | 125 | 92 | 474 |
| AG16 | 85 | 121 | 85 | 459 |
| ACY00 | 275 | 485 | 426 | 1949 |
| ACY03 | 269 | 474 | 424 | 1912 |
| AUY00 | 151 | 489 | 402 | 1941 |
| AUY03 | 151 | 485 | 402 | 1957 |
| C164 | 273 | 400 | 381 | 4048 |
| C165 | 301 | 420 | 397 | 4382 |
| HCA1 | 77 | 131 | 127 | 735 |
| HCA2 | 76 | 121 | 118 | 737 |
| HFUEA | 93 | 150 | 129 | 684 |
| HFULM | 92 | 148 | 129 | 680 |
| HFUMS | 91 | 146 | 134 | 670 |
| I01 | 155 | 222 | 244 | 2517 |
| I02 | 168 | 236 | 255 | 2396 |

### 3.4. Use of Different k-Values

With $k^2$-raster, we found that different $k$-values used in the structure would produce different bit rates and different access time. In general, for most of our testing images the $k$-value is at its optimal bit-rate level when it is between 4 and 9. The reason is that as the $k$-value increases, the height of the constructed tree becomes smaller. Therefore, the number of nodes in the tree will decrease and so will the size of the bitmaps *Lmax* and *Lmin* that need to be stored in the structure. Table 5 shows the bit rates of some of the testing images between $k = 2$ and $k = 20$. Additionally, experiments show that as the $k$-value becomes higher, the access time also becomes shorter, as can be seen in Table 6. As the $k$-value gets larger, the tree becomes shorter, thus making it faster to traverse from the top level to a lower level when searching for a particular node in the tree. As there is a trade-off between storage size and access time, for the experiments, the $k$-value that produces the lowest bit rate for the image was used.

For those who would like to know which $k$-value would give the best or close to the best rate, we recommend them to use a value of 6 as a general rule. This can be seen from Table 5 where the difference in the rate produced by this value and the one by the optimal $k$-value averages out to be only about 0.19 bpppb.

### 3.5. Use of Different Group Sizes

Tests were performed to see how the group size affects the predictive and differential methods. The group sizes were 2, 4, 8, 12, 16, 20, 24, 28 and 32. The results in Table 7 and Figure 13 show that for most images, they are at their optimal bit rates when the size is 4 or 8. The best bit-rate values are highlighted in red. For the range of group size tested, we can also see that except for the CRISM scenes (which consist of pixels with low spatial correlation, thus leading to inaccurate prediction), the bit rates for the predictor are always lower than the ones for differential encoding, irrespective of the group size.

For users who are interested in knowing which group size is the best to apply to the predictive and differential methods, a size of 4 is recommended for general use as the difference in bit rate produced by this group size and the one by the optimal group size averages out to be about 0.06 bpppb.

For the rest of the experiments, the optimal group size for each image was used to obtain the bit rate.

**Table 5.** Rates (bpppb) for different *k*-values for some of the testing images. The *k*-value with the lowest rate is in red.

| Hyperspectral Image | k = 2 | 3 | 4 | 5 | 6 | 7 | 8 | 9 | 10 | 11 | 12 | 13 | 14 | 15 | 16 | 17 | 18 | 19 | 20 |
|---|---|---|---|---|---|---|---|---|---|---|---|---|---|---|---|---|---|---|---|
| AG9 | 13.06 | 10.11 | 10.03 | 10.47 | 9.49 | 9.98 | 10.68 | 9.89 | 10.65 | - | 11.23 | 10.33 | 11.29 | 9.53 | 11.57 | 11.72 | 10.78 | 12.52 | 12.13 |
| AG16 | 12.72 | 9.78 | 9.66 | 10.11 | 9.12 | 9.57 | 10.32 | 9.51 | 10.29 | - | 10.82 | 9.98 | 10.86 | 9.17 | 11.11 | 11.28 | 10.32 | 12.07 | 11.68 |
| ACY00 | 12.34 | 10.20 | 9.76 | - | 9.61 | 9.91 | - | 9.69 | 9.83 | 9.87 | 9.95 | 10.24 | 10.20 | - | - | - | - | - | - |
| ACY03 | 11.81 | 9.87 | 9.56 | - | 9.42 | 9.71 | - | 9.50 | 9.65 | 9.70 | 9.76 | 10.01 | 9.98 | - | - | - | - | - | - |
| AUY00 | 15.31 | 12.93 | 12.20 | - | 12.08 | 12.35 | - | 11.92 | 12.11 | 12.13 | 12.17 | 12.52 | 12.43 | - | - | - | - | - | - |
| AUY03 | 15.03 | 12.60 | 12.00 | - | 11.90 | 12.20 | - | 11.74 | 11.93 | 11.94 | 12.00 | 12.34 | 12.25 | - | - | - | - | - | - |
| C164 | 12.60 | 10.42 | 10.17 | - | 10.08 | - | - | 10.34 | 10.20 | 10.76 | 10.48 | - | - | - | - | - | - | - | - |
| C165 | 12.84 | 10.67 | 10.48 | - | 10.37 | - | - | 10.54 | 10.51 | 10.79 | 11.03 | - | - | - | - | - | - | - | - |
| HCA1 | 10.79 | 9.41 | 8.85 | 8.45 | 8.74 | 9.36 | 8.20 | 8.51 | 8.68 | 8.85 | 8.88 | 8.92 | 9.21 | - | - | - | - | - | - |
| HCC | 9.43 | 8.12 | 7.79 | 7.41 | 7.75 | 8.40 | 7.38 | 7.67 | 7.85 | 8.06 | 8.12 | 8.26 | 8.56 | - | - | - | - | - | - |
| HFUEA | 8.82 | 7.80 | 7.30 | 7.24 | 7.41 | 8.07 | 6.84 | 7.25 | 7.43 | 7.66 | 7.68 | 7.71 | 8.07 | - | - | - | - | - | - |
| HFULM | 8.69 | 7.70 | 7.20 | 7.13 | 7.33 | 8.02 | 6.79 | 7.21 | 7.40 | 7.64 | 7.66 | 7.68 | 8.05 | - | - | - | - | - | - |
| I01 | 8.03 | - | 5.93 | - | - | 6.45 | - | - | - | - | - | - | - | - | 6.59 | 7.79 | 8.30 | 8.73 | 6.36 |
| I02 | 8.02 | - | 5.90 | - | - | 6.48 | - | - | - | - | - | - | - | - | 6.64 | 7.92 | 8.46 | 8.97 | 6.45 |

**Table 6.** Access time (ms) for different *k*-values for some of the testing images. The best access time is in red.

| Hyperspectral Image | Access Time (ms) | | | | | | | | | | | | | | | | | | | |
|---|---|---|---|---|---|---|---|---|---|---|---|---|---|---|---|---|---|---|---|
| | *k* = 2 | 3 | 4 | 5 | 6 | 7 | 8 | 9 | 10 | 11 | 12 | 13 | 14 | 15 | 16 | 17 | 18 | 19 | 20 |
| AG9 | 345 | 167 | 130 | 114 | 91 | 84 | 83 | 82 | 82 | - | 60 | 59 | 56 | 56 | 55 | 52 | 60 | 51 | <span style="color:red">47</span> |
| AG16 | 334 | 152 | 114 | 108 | 85 | 79 | 78 | 80 | 75 | - | 55 | 54 | 53 | 53 | 59 | 51 | 58 | 48 | <span style="color:red">45</span> |
| ACY00 | 3553 | 1085 | 573 | - | 291 | 225 | - | 152 | 133 | 125 | 114 | 110 | <span style="color:red">104</span> | - | - | - | - | - | - |
| ACY03 | 3521 | 1112 | 572 | - | 277 | 223 | - | 149 | 131 | 122 | 112 | 120 | <span style="color:red">102</span> | - | - | - | - | - | - |
| AUY00 | 3569 | 1135 | 592 | - | 292 | 228 | - | 153 | 133 | 126 | 115 | 113 | <span style="color:red">106</span> | - | - | - | - | - | - |
| AUY03 | 3559 | 1123 | 585 | - | 279 | 221 | - | 152 | 133 | 124 | 115 | 109 | <span style="color:red">103</span> | - | - | - | - | - | - |
| C164 | 2924 | 964 | 606 | - | 272 | - | - | 159 | 161 | 138 | <span style="color:red">131</span> | - | - | - | - | - | - | - | - |
| C165 | 3754 | 1017 | 555 | - | 290 | - | - | 178 | 156 | 152 | <span style="color:red">145</span> | - | - | - | - | - | - | - | - |
| HCA1 | 1179 | 384 | 213 | 154 | 124 | 106 | 80 | 78 | 69 | 71 | 62 | <span style="color:red">61</span> | 62 | - | - | - | - | - | - |
| HCC | 1203 | 406 | 233 | 172 | 139 | 123 | 95 | 94 | 86 | 85 | 83 | 79 | <span style="color:red">79</span> | - | - | - | - | - | - |
| HFUEA | 1409 | 465 | 262 | 184 | 148 | 127 | 93 | 95 | 89 | 84 | 77 | 76 | <span style="color:red">76</span> | - | - | - | - | - | - |
| HFULM | 1427 | 467 | 262 | 193 | 155 | 130 | 94 | 96 | 87 | 90 | <span style="color:red">79</span> | 81 | 80 | - | - | - | - | - | - |
| I01 | 999 | - | 779 | - | - | 679 | - | - | - | - | - | - | - | - | 610 | 709 | 715 | 728 | <span style="color:red">450</span> |
| I02 | 1047 | - | 759 | - | - | 698 | - | - | - | - | - | - | - | - | 651 | 746 | 746 | 730 | <span style="color:red">472</span> |

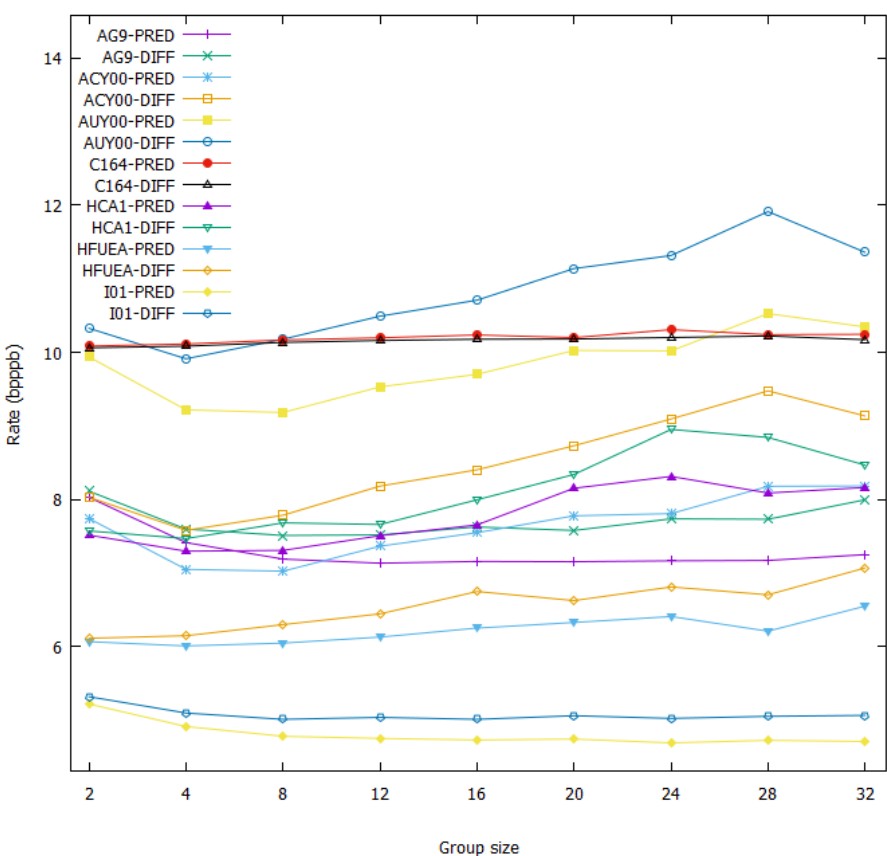

**Figure 13.** A rate (bpppb) comparison of different group sizes.

**Table 7.** A rate (bpppb) comparison of different group sizes using the predictive and the differential methods. The optimal values are highlighted in red.

| Hyperspectral Image | Group Size | | | | | | | | |
|---|---|---|---|---|---|---|---|---|---|
| | **2** | **4** | **8** | **12** | **16** | **20** | **24** | **28** | **32** |
| AG9-Pred | 8.03 | 7.41 | 7.19 | 7.13 | 7.16 | 7.15 | 7.17 | 7.17 | 7.25 |
| AG9-Diff | 8.11 | 7.60 | 7.51 | 7.52 | 7.63 | 7.58 | 7.74 | 7.73 | 8.00 |
| ACY00-Pred | 7.74 | 7.05 | 7.03 | 7.36 | 7.55 | 7.78 | 7.81 | 8.18 | 8.18 |
| ACY00-Diff | 8.03 | 7.58 | 7.79 | 8.18 | 8.40 | 8.73 | 9.09 | 9.48 | 9.13 |
| AUY00-Pred | 9.94 | 9.22 | 9.18 | 9.53 | 9.70 | 10.03 | 10.02 | 10.53 | 10.34 |
| AUY00-Diff | 10.33 | 9.91 | 10.18 | 10.49 | 10.71 | 11.14 | 11.32 | 11.91 | 11.36 |
| C164-Pred | 10.08 | 10.11 | 10.17 | 10.20 | 10.23 | 10.20 | 10.31 | 10.24 | 10.24 |
| C164-Diff | 10.06 | 10.08 | 10.13 | 10.16 | 10.18 | 10.20 | 10.31 | 10.22 | 10.17 |
| HCA1-Pred | 7.51 | 7.30 | 7.31 | 7.51 | 7.65 | 8.15 | 8.31 | 8.09 | 8.16 |
| HCA1-Diff | 7.57 | 7.47 | 7.68 | 7.66 | 8.00 | 8.34 | 8.95 | 8.84 | 8.47 |
| HFUEA-Pred | 6.07 | 6.01 | 6.05 | 6.13 | 6.25 | 6.33 | 6.41 | 6.21 | 6.55 |
| HFUEA-Diff | 6.11 | 6.15 | 6.30 | 6.44 | 6.75 | 6.63 | 6.81 | 6.70 | 7.07 |
| I01-Pred | 5.22 | 4.91 | 4.78 | 4.75 | 4.73 | 4.74 | 4.69 | 4.73 | 4.71 |
| I01-Diff | 5.32 | 5.10 | 5.01 | 5.04 | 5.01 | 5.06 | 5.02 | 5.05 | 5.06 |

### 3.6. Predictive and Differential Methods

The proposed differential and predictive methods were used to transform these images into data with lower bit rates. They were then used as input to $k^2$-raster to further reduce their bit rates. Their performance was compared together with Reversible Haar Transform at levels 1 and 5, and the results are presented in Table 8. Figure 14 shows the entropy comparison of Yellowstone03 using differential and predictive methods while Figure 15 shows the bit rate comparison between the two

methods. Both show us that the proposed algorithm has brought benefits by lowering the entropy and the bit rates. The data for reference bands are left out of the plots so that the reader can have a clearer overall picture of the bit rate comparison.

Compared to other methods, the predictive method outperforms others, with the exception of Reversible Haar Transform level 5. However, it should be noted that while the predictive and differential methods require only two pixels (reference pixel and current pixel) to perform the reverse transformation, it would be a much more involved process to decode data using Reversible Haar Transform at a higher level. The experiments show that for all the testing images, the predictive method in almost all bands perform better than the differential method. This can be explained by the fact that in predictive encoding the values of $\alpha$ and $\beta$ in Equation (1) take into account not only the spectral correlation, but also the spatial correlation between the pixels in the bands when determining the prediction values. This is not the case with differential encoding whose values are only taken from the spectral correlation.

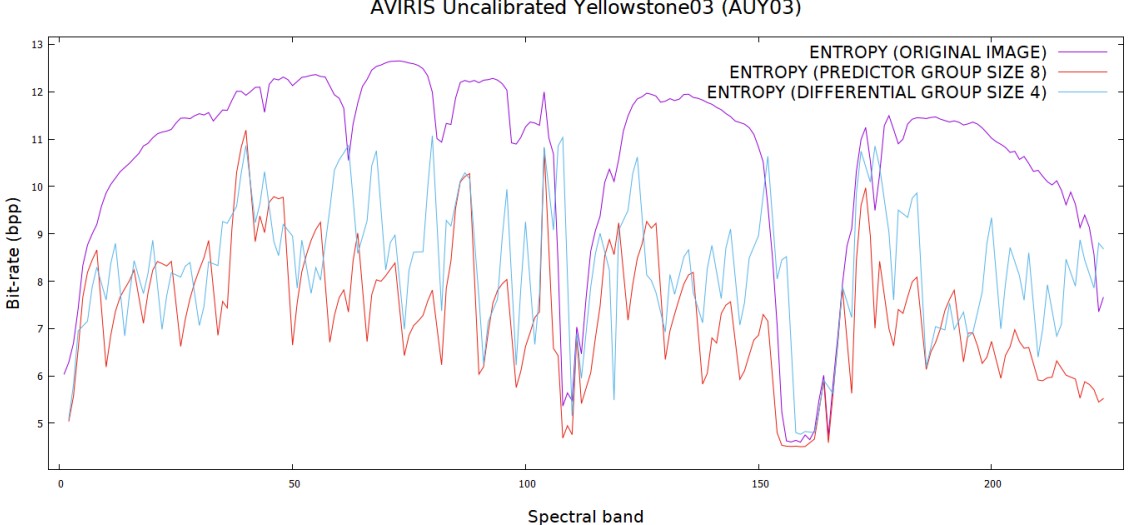

**Figure 14.** An entropy comparison of Yellowstone03 using differential and predictive methods. Data for reference bands are not included.

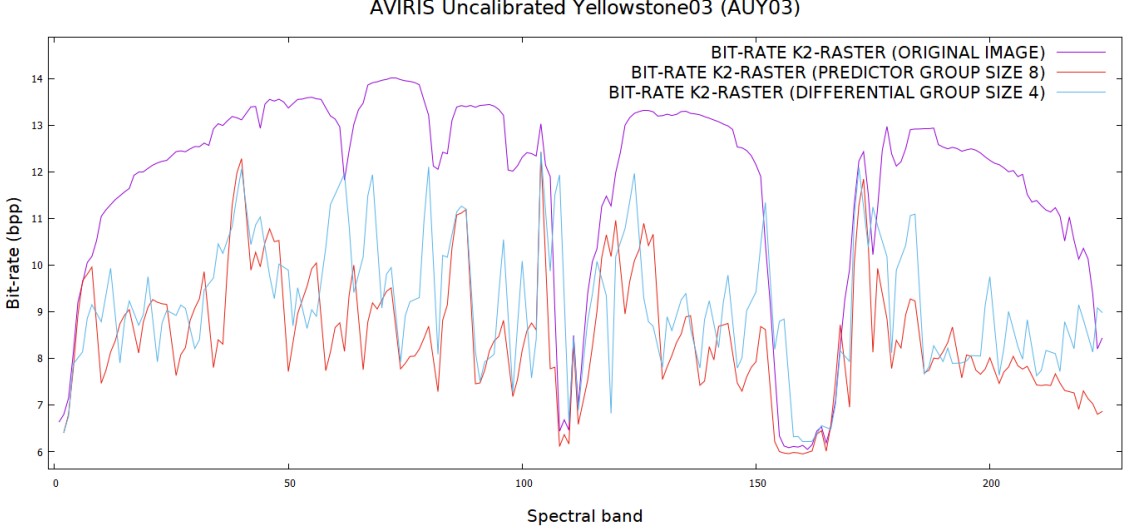

**Figure 15.** A bit rate comparison of Yellowstone03 using differential and predictive methods on $k^2$-raster. Data for reference bands are not included.

**Table 8.** A rate (bpppb) comparison using different transformed methods: predictor, differential, reversible Haar level 1 and reversible Haar level 5 on $k^2$-raster. The optimal values are highlighted in red.

| Hyperspectral Image | Transformation Type | | | | |
|---|---|---|---|---|---|
| | Without Transformation | Predictor | Differential | Reversible Haar (Level 1) | Reversible Haar (Level 5) |
| AG9 | 9.49 | 6.76 | 7.52 | 8.10 | 6.83 |
| AG16 | 9.12 | 6.63 | 7.29 | 7.81 | 6.60 |
| ACY00 | 9.63 | 6.87 | 7.79 | 8.01 | 7.00 |
| ACY03 | 9.44 | 6.72 | 7.65 | 7.86 | 6.87 |
| AUY00 | 11.92 | 9.04 | 10.04 | 10.33 | 9.35 |
| AUY03 | 11.74 | 8.87 | 9.91 | 10.18 | 9.23 |
| C164 | 10.08 | 10.02 | 10.06 | 10.01 | 9.83 |
| C165 | 10.37 | 10.33 | 10.37 | 10.33 | 10.16 |
| HCA1 | 8.20 | 7.47 | 7.47 | 7.37 | 7.05 |
| HCC | 7.38 | 7.50 | 7.50 | 6.71 | 6.54 |
| HFUEA | 6.84 | 5.99 | 6.15 | 7.12 | 6.75 |
| HFULM | 6.79 | 6.06 | 6.18 | 7.14 | 6.83 |
| I01 | 5.93 | 4.69 | 5.01 | 5.26 | 4.54 |
| I02 | 5.90 | 4.75 | 5.03 | 5.26 | 4.57 |

## 4. Conclusions

In this work, we have shown that using $k^2$-raster structure can help reduce the bit rates of a hyperspectral image. It also provides easy access to its elements without the need for initial full decompression. The predictive and differential methods can be applied to further reduce the rates. We performed experiments that showed that if the image data are first converted by either a predictive method or a differential method, we can gain more reduction in bit rates, thus making the storage capacity or the transmission volume of the data even smaller. The results of the experiments verified that the predictor indeed gives a better reduction in bit rates than the differential encoder and is preferred to be used for hyperspectral images.

For future work, we are interested in exploring the possibility of modifying the elements in a $k^2$-raster. This investigation is based on the dynamic structure, $dk^2$-tree, as discussed in the papers by de Bernardo et al. [29,30]. Additionally, we would like to improve on the variable-length encoding which is currently in use with $k^2$-raster, and hope to further reduce the size of the structure [23,24].

**Author Contributions:** conceptualization, K.C., D.E.O.T., I.B. and J.S.-S.; methodology, K.C., D.E.O.T., I.B. and J.S.-S.; software, K.C.; validation, K.C., I.B. and J.S.-S.; formal analysis, K.C., D.E.O.T., I.B. and J.S.-S.; investigation, K.C., D.E.O.T., I.B. and J.S.-S.; resources, K.C., D.E.O.T., I.B. and J.S.-S.; data curation, K.C., I.B. and J.S.-S.; writing—original draft preparation, K.C., I.B. and J.S.-S.; writing—review and editing, K.C., I.B. and J.S.-S.; visualization, K.C., I.B. and J.S.-S.; supervision, I.B. and J.S.-S.; project administration, I.B. and J.S.-S.; funding acquisition, I.B. and J.S.-S.

**Funding:** This research was funded by the Spanish Ministry of Economy and Competitiveness and the European Regional Development Fund under grants RTI2018-095287-B-I00 and TIN2015-71126-R (MINECO/FEDER, UE) and BES-2016-078369 (Programa Formación de Personal Investigador), and by the Catalan Government under grant 2017SGR-463.

**Acknowledgments:** The authors would like to thank Magli et al. for the M-CALIC software that they provided us in order to perform some of the experiments in this research work.

**Conflicts of Interest:** The authors declare no conflict of interest.

**Abbreviations**

The following abbreviations are used in this manuscript:

| | |
|---|---|
| AIRS | Atmospheric Infrared Sounder |
| AVIRIS | Airborne Visible InfraRed Imaging Spectrometer |
| CALIC | Context Adaptive Lossless Image Compression |
| CCSDS | Consultative Committee for Space Data Systems |
| CRISM | Compact Reconnaissance Imaging Spectrometer for Mars |
| DACs | Directly Addressable Codes |
| IASI | Infrared Atmospheric Sounding Interferometer |
| JPEG 2000 | Joint Photographic Experts Group 2000 |
| KLT | Karhunen–Loève Theorem |
| LOUDS | Level-Order Unary Degree Sequence |
| MDPI | Multidisciplinary Digital Publishing Institute |
| PCA | Principal Component Analysis |
| SOAP | Short Oligonucleotide Analysis Package |

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
