# Peer review of "Using Predictive and Differential Methods with K2-Raster Compact Data Structure for Hyperspectral Image Lossless Compression†"

_remotesensing, doi:10.3390/rs11212461_

Round 1

Reviewer 1 Report

The proposed paper is potentially interesting, but it needs some clarifications and improvements, as reported in the following list of remarks:

1) A possible utilization for an on-board compression of the acquired data of this procedure is not discussed. In particular, possible advantages, if they were, of this procedure in such an environment with respect to other compression techniques should be evidenced.

2) The authors state the their prediction is based on some concepts described in the 3D-CALIC paper. Is it possible to insert this method in the comparisons?

3) Most of the concepts (and similar figures) developed in this paper can be also found in the paper [16]. However, the differences of the proposed procedure and that of [16] are not clearly underlined. What is the gain in performances on the considered data sets between the proposed method and that of [16]?

4) In Equation (1) the coefficients \alpha and \beta are generic real numbers, but in Equations from (2) to (5) the coefficients \alpha and \beta are optimally computed. Therefore, they should be evidenced with a hat, for example.

Reviewer 2 Report

The present paper submitted by Chow et al. describes a new methodology to compress hyperspectral images.

The paper is clear and well written. However, no clear discussion about image quality after compression is provided to the reader. This should be added to the text to have a clear idea of the potential interest of the described method.

Reviewer 3 Report

This manuscript, they proposed a lossless coder for real-time processing and compression of hyperspectral images. They said that, the advantage of using such a data structure is its compactness, with a size that is comparable to that produced by some classical compression algorithms and yet still providing direct access to its content for query without any need for decompression. Experimental results that show that the higher rates reduction than differential encoding.

Some comments are as follows:

1) Why do authors choose hyperspectral images as the object of interest for their algorithms?

2) What does the hyperspectral image compression mean in practice?

3) Need to clarify the computational complexity of the algorithm?

4) Need to convince readers by using the method of performing hyperspectral images before compression and after decompressing by visualization.

5) It is necessary to cite and discuss more recent studies related to the image compression and the hyperspectral image compression.

6) Experimental results need to calculate the time of encoding and decoding on hyperspectral image data.

7) In my opinion, the experimental results presented in Figure 14 and Figure 15 do not indicate the effectiveness of the proposed algorithm.

Reviewer 4 Report

I have found this paper quite clear and well written.
The Authors have provided a well-structured exposition of their material.
The content is described with a quite sufficient amount of details to understand the topic, techniques and results.
The analysis provided and corresponding results are quite appropriate to the text and its content.
The list of references to the literature related to the field is also appropriate.

Overall, the content is quite original. But the argument and analysis needs improvement as they lack details and developments in some places.

Despite the lack of leadership in the results in terms of rate comparison (bpppb) with other compression techniques (as shown in Figure 12), the proposed compression scheme exhibits a good behaviour (close to best achievable in the comparison performed) with certainly additional benefits (direct access to the elements without decompression).

1) Now, my main concern is about the optimal determination of k-values as well as the group size for prediction and differential encoding. Trends are analyzed correctly although this could probably even more developed.

2) Given a multidimensional dataset to process, how to do to find the best (optimal) value of these two parameters (in terms of bpppb and/or access time) ?
How to find the right value other than by choosing a value from a suggested range of values proved adequate on a large but ultimately limited experimental dataset?

3) Can the practitioner make an optimal tuning for both parameters at the same time?
Or is he necessarily forced to satisfy a compromise that he has to resolve by himself?
Then, what are appropriate recommendations or key steps to help him solving this task?

Typo(s):
Please define bpppb

Round 2

Reviewer 1 Report

The authors have addressed all my previous remarks, even if a statement about on-board compression should be added to the paper, by especially considering that several references concern on-board compression and this paper is an extended version of a work presented in a workshop on on-board data compression.

Moreover, some minor typos can be still found on this paper, and in particular:

a) Page 2, line 31: I found hypersepectral instead of hyperspectral;

b) Page 2, line 59: latest is in lowercase;

c) Section 3 is repeated three times. Sections 2.1 and 2.2 at page 10 should be 2.8 and 2.9. Please check the progressive numeration of the Sections.

d) Page 14, line 330: \alpha and \beta should be with hat; alternatively, the statement should be: "the optimal values of \alpha and \beta".

Reviewer 2 Report

The authors answered my concerns. The paper may be published.

Reviewer 3 Report

I have no comments
